# Assessment of Muscle Coordination Changes Caused by the Use of an Occupational Passive Lumbar Exoskeleton in Laboratory Conditions

**DOI:** 10.3390/s23249631

**Published:** 2023-12-05

**Authors:** Sofía Iranzo, Juan-Manuel Belda-Lois, Jose Luis Martinez-de-Juan, Gema Prats-Boluda

**Affiliations:** 1Instituto de Biomecánica de Valencia, Universitat Politècnica de València, 46022 Valencia, Spain; sofia.iranzo@ibv.org (S.I.); juanma.belda@ibv.org (J.-M.B.-L.); 2Centro de Investigación e Innovación en Bioingeniería (Ci2B), Universitat Politècnica de València, 46022 Valencia, Spain; jlmartinez@eln.upv.es

**Keywords:** muscle interaction, passive industrial exoskeleton, electromyography, mutual information, multivariate sample entropy, Granger causality

## Abstract

The introduction of exoskeletons in industry has focused on improving worker safety. Exoskeletons have the objective of decreasing the risk of injury or fatigue when performing physically demanding tasks. Exoskeletons’ effect on the muscles is one of the most common focuses of their assessment. The present study aimed to analyze the muscle interactions generated during load-handling tasks in laboratory conditions with and without a passive lumbar exoskeleton. The electromyographic data of the muscles involved in the task were recorded from twelve participants performing load-handling tasks. The correlation coefficient, coherence coefficient, mutual information, and multivariate sample entropy were calculated to determine if there were significant differences in muscle interactions between the two test conditions. The results showed that muscle coordination was affected by the use of the exoskeleton. In some cases, the exoskeleton prevented changes in muscle coordination throughout the execution of the task, suggesting a more stable strategy. Additionally, according to the directed Granger causality, a trend of increasing bottom-up activation was found throughout the task when the participant was not using the exoskeleton. Among the different variables analyzed for coordination, the most sensitive to changes was the multivariate sample entropy.

## 1. Introduction

Exoskeletons have become a popular technology in recent years due to their potential to enhance human performance and protect against musculoskeletal injuries. These wearable devices are designed to improve or replace the function of the human musculoskeletal system by providing external support, assistance, or resistance. However, the use of exoskeletons could also have additional consequences, such as changes in muscle activation patterns, that are yet to be studied and described.

In recent years, numerous studies have evaluated the impact of industrial exoskeletons on ergonomics in the workplace [1,2,3,4,5]. These studies have generally focused on assessing the effects of exoskeletons on the physical strain and discomfort experienced by workers during tasks that involve repetitive or heavy lifting, bending, or reaching. The results of these studies have been controversial, with some suggesting that exoskeletons can effectively reduce muscle activity, joint forces, and fatigue, while others have indicated no significant effect or even negative outcomes [6]. Previous works related to studying the effects of industrial passive exoskeletons on users have included the assessment of muscles through EMG [7,8]. These studies addressed the quantification of muscular activity and fatigue, focusing only on the effects on each muscle individually.

Studies on this topic start with the identification of the muscles affected by the use of the exoskeleton. In the case of a passive lumbar exoskeleton, the lumbar muscle is selected as the objective muscle of the device, as it should receive the most benefit. The authors of [9,10,11] all found a reduction in the lumbar activity when assessing the same exoskeleton, but this reduction was also observed in studies examining different back support exoskeletons to that used in the present study [8,12,13,14]. Furthermore, reductions in lumbar fatigue have also been found in numerous works [9,15,16]. On the other hand, the quadriceps muscle has also been studied, as it is potentially affected by compensating for reduced lumbar muscle activity. Iranzo et al. [9] found reduced activity but no significant changes in fatigue [9,17].

The interaction between exoskeletons and the human body is complex, and understanding the underlying mechanisms is critical for optimizing the design and effectiveness of these devices. One important factor to consider is the interaction between muscles, which refers to the coordinated activation of different muscles to produce a desired movement or force [18]. Muscle interactions play a crucial role in human movement and could be affected by exoskeleton use.

One of the most common ways to study muscle couplings in the literature consists in calculating certain parameters, such as the correlation coefficient, coherence coefficient, mutual information, and multivariate sample entropy [19,20]. These variables have been proposed in the literature to evaluate muscular coupling for smooth and striated muscles, though considering different muscles and applications to those studied in the present work.

The correlation coefficient can reflect the linear correlation between two signals in the time domain. It has been used in the literature to characterize pairs of uterine muscle synchronizations before labor [19,21], with a significant increase in its value as the delivery approached. Furthermore, King [22] used it to find correlations during movement compared to static poses, also obtaining higher values in dynamic versus static conditions. The coherence coefficient can also reflect the linear correlations between time series in the frequency domain, and it is widely used in the literature. De Marchis et al. [23] calculated this variable to study the intermuscular synchronization in a free pedaling task, finding peak values of coherence in the soleous and gastrocneminus medialis pair of muscles. Coherence was also assessed for labor prediction in [21], with greater values obtained as delivery approached. Coherence was computed to study post-stroke muscle interactions in the deltoid and triceps muscles in [24], and the values of coherence were lower for patients than for the control.

The mutual information coefficient (MI) is a general method for detecting linear and nonlinear statistical dependencies between time series. This variable was also assessed in the aforementioned work on uterine muscle synchronizations [19,21], with higher values of MI as the delivery advanced. Furthermore, the authors of Wu et al. [25] proposed a methodology based on MI to analyze intermuscular coupling during the movement of the upper limbs, obtaining higher values for the triceps brachii and posterior deltoid pair compared to static states. MI was also utilized to measure the inter-muscular coupling between the biceps and triceps with aging [26], with decreased values of MI obtained as aging progressed. The authors of Svendsen et al. [27] used MI to reflect the inter-muscular coupling of four forearm muscles during static and dynamic tracking tasks, with greater values obtained for static states.

The multivariate sample entropy (MSE) measures the structural complexity of real-world multichannel data by examining nonlinear correlations within and between channels. It provides a robust relative complexity measurement for multivariate data and has been validated on real-world multivariate gait, physiological, and wind data [28]. It has also been used to study uterine muscles during delivery progression [21], demonstrating very low values with the approach of delivery. Muscle interactions between dynamic and resting states were also addressed by MSE computation in [22], finding lower values and therefore stronger couplings in the external oblique and transverse pair of muscles. In a different study, the authors calculated the MSE for pairs of muscles under different conditions of gait and running speed [29].

Finally, besides the approach of studying the described parameters to find interactions between pairs, the causal relationship between pairs is of great relevance. The conditional Granger causality (CG-Causality) analyzes the directed functional coordination between pairs of muscles. The authors of Ye-Lin et al. [20] used the CG-Causality from surface electromyography signals to examine the directed functional coordination of various swallowing muscles during the ingestion of different liquids in both healthy and dysphagic subjects. The authors of Zhou et al. [30] implemented a method of PCA-based CG-Causality to detect brain network connectivity. As a step further, the directed conditional Granger causality (DCG-Causality) parameter adds information about the direction of the causality. This calculation was performed in the study of uterine muscle synchronization to determine the direction in which the signals propagated, revealing that the majority of signals propagated downward to expulse the fetus [1].

The present paper aimed to investigate the couplings and synchronization between pairs of muscles when using an exoskeleton. To achieve this goal, we conducted EMG recordings, processed them, and obtained the aforementioned parameters to assess correlations and couplings between pairs of signals: the correlation coefficient, coherence coefficient, mutual information, and multivariate sample entropy. Additionally, we used the DCG-Causality parameter to examine the direction of causality. In summary, this study sought to contribute to a better understanding of the complex interactions between exoskeletons and the human body, providing insights into how to optimize the design and use of these devices for various applications.

## 2. Materials and Methods

To participate in the study, individuals of both genders had to meet following criteria: being between 30 to 45 years old and having a body mass index (BMI) within the range of 18.5 kg/m${}^{2}$ to 25.5 kg/m${}^{2}$. Individuals with a history of musculoskeletal lesions or respiratory or cardiovascular pathologies were excluded. Specifically, the study involved 8 volunteers, consisting of 4 women and 4 men, who visited the Instituto de Biomecánica de Valencia (IBV) facilities and provided written consent for the use and publication of their data. The average weight and height of the participants were 67.9±7.8 kg and 175.6±4.6 cm, respectively, with standard deviations indicated.

### 2.1. Setup Design

The task design aimed to replicate common manual handling tasks in industrial and warehouse settings, which typically involve a high physical load and adheres to ergonomic requirements. Although the designed tasks may not have encompassed all possible postures involved in carrying heavy objects, they simulated a depalletizing job that involves musculoskeletal risks from forced postures. The selected tasks recreated a stationary workstation with limited dynamic movements that could necessitate minimal support, as occurs in workstations that use passive exoskeletons. The task design was based on ergonomic risk factors, and all the details can be found in our previous work [9].

In summary, the tasks consisted of depalletizing a block of four rows of four boxes. Figure 1 shows the type of box over the destination table and a schematic drawing in white of the initial configuration of the 16-box pallet.

To ensure consistency across users, a predetermined pattern was followed when moving the boxes from the pallet to the destination. The 16 boxes were numbered in sequence from the top row (boxes 1 to 4) to the bottom row (boxes 13 to 16). The users performed the depalletizing task six times. The first three repetitions were performed without the exoskeleton, handling weights of 7 kg, 8 kg, and 9 kg, respectively. The second three were performed with the exoskeleton, handling 7 kg, 8 kg, and 9 kg weights in each repetition. Therefore, under the conditions of no exoskeleton, 48 boxes were moved in total, and then 48 boxes were moved with the exoskeleton. Finally, the rhythm was indicated by a metronome sound every 6 s (frequency of manipulation).

The recording protocol included a 10-min break between sessions with and without the exoskeleton. This break, exceeding by 6 times the exercise execution time of approximately 60 s, allowed for recovery in case the subject experienced fatigue during the exercise without the exoskeleton.

### 2.2. Equipment

The exoskeleton used was the commercial passive lumbar Laevo^TM^ V2 exoskeleton [10,31,32].

EMG signals were measured using a Noraxon wireless electromyography system (UltiumTM EMG) to monitor the muscular activity of the right side muscles: erector spinae (LUMB), gluteus medius (GLUT), quadriceps femoris (QUAD), and semitendinosus (SEMI). The signals were sampled at 2000 Hz. A clinical evaluator followed the SENIAM guidelines [33] to place the bipolar electrodes.

The Xsens^TM^ MVN Analyze system in whole-body configuration was used for motion capture; these data were collected to track the postures of the users in order to perform the segmentation of the EMG signals at the desired positions. Both systems, EMG and MoCap, were synchronized using the Noraxon Myosync channel.

### 2.3. Data Analysis

#### 2.3.1. Signal Preprocessing

The EMG signal pre-processing consisted of two stages. First, a filtering stage was implemented to clean and prepare the signals. Once the EMG signals were obtained, a zero-phase bandpass Butterworth filter of order 10 was used for pre-processing. The cut-off frequencies of 20 and 200 Hz were applied to suppress movement noise and limit the study’s bandwidth.

Secondly, a segmentation task for selecting the fragments of muscular signal activation when the user was holding the box, from lifting to downloading, was carried out, which was the same for all muscle channels. In total, 48 fragments (three exercises of 16 boxes each) were selected for each muscle in both the with- and without-exoskeleton conditions. The extended details of the segmentation methodology, which was based on calculating the envelope of the EMG signals in the four muscles by rectifying and smoothing with a 4 Hz low-pass filter, can be found in a previous work [9].

#### 2.3.2. Feature Extraction

The interaction among the erector spinae, gluteus medius, quadriceps femoris, and semitendinosus muscles was analyzed by computing the correlations and information-theory-derived parameters in pairs of EMG signals. Specifically, a set of five parameters was calculated from each of the 48 common fragments of each muscle pair in both conditions, with and without the exoskeleton. The first four corresponded to the non-directional parameters of the correlation coefficient, coherence coefficient, mutual information, and multivariate sample entropy.

The correlation coefficient (CORR) expressed the linear correlation between a pair of EMG signals in the time domain. The CC ranged from −1 to 1 and equaled 0 when there was no linear correlation between signals. With the use of the exoskeleton, the values could be expected to be higher in specific muscle pairs because, due to the design of the exoskeleton, some muscles were forced to be coupled.

The coherence quantifies the linear correlation between a pair of signals in the frequency domain, being an extension of Pearson’s correlation coefficient in the frequency domain. In this paper, the maximum value of this function was considered (coherence coefficient, COH). The values of the COH ranged from 0 to 1, and the closer the coefficient was to 1, the more linear the relation between both signals. The closer to 0, the less closely related the signals were. As in the case of the CORR, with the use of the exoskeleton, the values could be expected to be higher.

Mutual information (MI) measures the amount of information that one random variable contributes to another variable [34,35]. The higher the correlation between the pair of EMG signals, the higher the value of the MI, which was zero when the pair of signals were statistically independent. The behavior of this parameter was expected to be comparable to the CORR and COH parameters, i.e., the higher the MI the higher the muscle coupling.

Multivariate sample entropy characterizes the likelihood that similar patterns in a time series will remain similar over time among multichannel data [36]. In this case, in contrast to the CORR, COHM, and MI, the MSE values were expected to be lower for a higher degree of coupling.

Furthermore, we computed the directional conditional Granger causality (CG-Causality). This is defined as the power of prediction that the past of a signal Y has for a signal X, in addition to the prediction of X made by its own past and the past of a conditioning variable Z [37]. In the case of EMG signals, CG-Causality allowed us to detect interactions between muscles and uncommon causal influences [37]. This parameter expresses directionality, and so the effect of the exoskeleton depended on the pair of muscles studied. The results revealed the degree to which the device made one muscle the “director” of another, in contrast to their relationship in the absence of the exoskeleton.

Figure 2 shows an example of the signals and parameters of a specific user. It also depicts the corresponding EMG coupling parameters of this muscle pair in the conditions with and without the exoskeleton. The four subplots at the top contain the EMG signal fragments that belong to each of the raised boxes. In the four subplots, the two pairs of measurements for the lumbar and gluteus muscles can be observed, with each pair containing a signal in red, indicating recordings in which the exoskeleton was used, and one in blue, for recordings without an exoskeleton.

In the last row, the series of plots correspond to the parameter values calculated for each of the fragments of the muscle pair. The trend lines in red correspond to the condition with the exo, and those in blue the condition without the exo. With the purpose of analyzing the parameters obtained from all users, a mixed model was built to calculate the trend lines for the whole set. In the following section, the mixed model is described, together with the post hoc analysis carried out to obtain the slopes and intercepts for each pair of muscles.

#### 2.3.3. Mixed Model

The data treatment and posterior statistical analysis performed had the objective of finding evidence of changes in the values and patterns of muscle synergies between the conditions with and without the exoskeleton. The interactions per fragment were characterized by the feature calculated for each muscle pair, as shown in the first row of Figure 2.

The main hypothesis was that the exoskeleton could affect muscle couplings, and muscle couplings could also be affected by fatigue components. For this reason, the data were analyzed to find the significant differences (*p* < 0.05) in the values of the parameters under each condition, as well as the significant differences in the evolution of the parameters throughout the exercise and how the fatigue that appeared over time affected the possible changes in the couplings.

The fatigue component was reflected in the trend of the parameters; therefore, the order, understood as the position of the box (1 to 48), was considered as a numerical factor. Furthermore, the interaction between the use of the exoskeleton and the order was considered in the model. The user was introduced as a random factor in Model (Equation 1).
(1)y(feature,musclepair)~exo∗order+(1|user)

This calculus was carried out for each y(feature,musclepair), five features, and six muscle combinations. The mixed model was built in R using the R package lme4 [38].

#### 2.3.4. Statistical Analysis

A post hoc analysis was performed and adjusted by the Holm method using the phia package in R [39,40].

Once the cases with significant differences (*p* < 0.05) were identified, the values of the slopes and intercepts were extracted from the model. With the slopes and intercepts, it was possible to appreciate whether the trend of the parameter values was increasing or decreasing for each case and draw comparisons between conditions with and without the exoskeleton.

## 3. Results

Figure 3 shows a matrix of representations of the coupling parameter trends for the conditions with and without the exoskeleton. The graphical representation of the trends was built using the obtained slopes and intercepts, with x-axis values from 1 to 48 representing each of the 48 order positions of each box handled (16 7 kg boxes, followed by 16 8 kg boxes and 16 9 kg boxes) under each condition. The slopes and intercepts used to build the lines were obtained from the mixed model (Equation 1). The blue dashed lines represent the condition without the exoskeleton, and the red solid lines represent the condition with the exoskeleton. The columns indicate each of the four coupling parameters: the correlation coefficient (CORR), coherence coefficient (COH), mutual information (MI), and multivariate sample entropy (MSE). The rows contain each of the six muscle combinations: semitendinosus–quadriceps (SEMI-QUAD), lumbar–quadriceps (LUMB-QUAD), quadriceps–gluteus (QUAD-GLUT), gluteus–lumbar (GLUT-LUMB), lumbar–semitendinosus (LUMB-SEMI), and gluteus–semitendinosus (GLUT-SSEMI).

In this matrix, the plots without grey shadowing are the ones that show significant differences. Over each plot in the Figure 3, there is a tag with the type of significant differences found. “V” corresponds to differences found between the conditions with and without the exoskeleton, meaning that the condition produced a significant change in the values of the interactions found. “S” corresponds to differences found in the slope between the conditions, meaning that the conditions significantly changed the way that the coupling evolved throughout the whole exercise (48 boxes). “V & S” corresponds to significant differences for both the value and the slope. Table 1 includes the *p*-values corresponding to each pair of muscles and variable.

For all the muscle pairs, significant differences were found between the conditions in two or more parameters.

When observing the MSE and MI parameters in all cases, the evolution remained nearly constant across all positions and boxes when using the exoskeleton. Conversely, in all cases, an upward trend was observed when the task was carried out without the exoskeleton. This observation holds significant implications for both parameters, particularly for the GLUT-LUMB, SEMI-QUAD, LUMB-QUAD, and QUAD-GLUT pairs, which considered the interaction of the quadriceps with other muscles in terms of the MSE.

The mutual information (MI) consistently exhibited lower values throughout almost all of the exercise when no exoskeleton was used. Conversely, the MSE showed exactly the opposite trend, and this distinction was more pronounced. These results aligned with the theoretical expectations associated with the variables. Specifically, higher MI values indicated stronger coupling between the muscle pairs, whereas lower MSE values signified a higher degree of coupling.

It was also noticeable that the MI values for the LUMB-SEMI muscle pair when the exoskeleton was used reinforced the mutual information of all boxes for this muscle pair, with very subtle variations throughout the exercise. This could have been due to “release” efforts from the lumbar to the semitendinosus muscle. The MI for the GS behaved in a similar way. For the GLUT-LUMB pair, it is remarkable that the use of the exoskeleton tended to maintain a constant coupling between muscles.

The correlation and the coherence parameters showed statistically significant differences in several combinations of muscles, although none of the pairs had the same results. In the case of the CORR values, the differences were significant, and for all significant pairs, the values between conditions agreed with those obtained for the MI parameter. For the LUMB-QUAD and LUMB-SEMI pairs, the values of the CORR were higher when wearing the exoskeleton, which indicated a higher degree of coupling in terms of linear correlation between the LUMB and SEMI and QUAD muscles when using the exoskeleton, perhaps due to the transfer of efforts that the exoskeleton caused. In the case of the GLUT-SEMI pair, the exoskeleton kept the coupling almost constant in terms of linear correlation between the GLUT and SEMI muscles, with similar values to those at the beginning of the exercise without the exoskeleton. This effect was also present in the COH parameter for the pairs QUAD-GLUT and GLUT-LUMB. Moreover, a significant difference was found for the slope of the COH parameter for the GLUT-LUMB pair, in agreement with the differences found in the MI and MSE for the same pair of muscles, which were almost constant when wearing the exoskeleton.

Figure 4 presents the results of the mixed model for the directional conditional Granger causality feature. This parameter is presented separately due to its characteristic of expressing directionality, which is not applicable to the other parameters. In this case, each combination of muscles is indicated in each plot, and the type of significant difference is annotated in the plot where it was found, in the same way as in Figure 3. Table 2 includes the *p*-values of the significant differences found in Figure 3. Here, the dashed line represents the condition without the exoskeleton, and the solid line represents the condition with the exoskeleton.

For this variable, there were statistically significant differences in the slope for the SEMI-QUAD and GLUT-LUMB pairs, i.e., in the way that the parameter of the DCG-Causality evolved throughout the exercise. In both cases, the values of the condition with the exoskeleton evolved in a more steady way (as observed for the MSE, MI, and CORR), and so the values of SEMI-QUAD minus QUAD-SEMI and GLUT-LUMB minus LUMB-GLUT were more constant. In the case of the no-exoskeleton condition, these values changed to either a negative value (SEMI-QUAD minus QUAD-SEMI) or positive value (GLUT-LUMB minus LUMB-GLUT). The Granger causality values presented a downward trend with negative values for GLUT-QUAD minus QUAD-GLUT and an upward trend with positive values for GLUT-LUMB minus LUMB-GLUT. This indicated that the exoskeleton tended to soften the evolution of the muscle interactions during the exercise.

## 4. Discussion

Although many studies have been reported in the last few years (Google Scholar produced 1340 references on 28 November 2023 for the keywords Exoskeleton, EMG, and Workplace), most of them were related to the comparison of individual muscle activation with and without an exoskeleton [41,42,43].

In this study, we aimed to assess how the use of the exoskeleton affected the coupling and synchronization (with and without directionality) between the pairs of muscles involved. A number of findings on the effects of using the exoskeleton were derived from the analysis.

Firstly, fatigue was evident in the MSE, and it was higher when there was no exoskeleton because the entropy increased, indicating that the signal was less predictable over time, and the COH decreased. This adds context to previous findings on the effects of fatigue in the use of passive lumbar exoskeletons [9]. The impact in fatigue has been addressed previously in several studies [44,45], mostly as an extrapolation on the effects derived from the reduction of the EMG activity with the use of an exoskeleton. In our study, fatigue without the exoskeleton also manifested in the increased predictability of the gluteus relative to the erector spinae and the increased predictability of gluteus activation relative to quadriceps activation, indicating that as fatigue increased the activation was increasingly bottom-up.

However, the main purpose of this contribution was to understand how muscle coordination was affected by the use of the exoskeleton, and how this coordination evolved over time while wearing the exoskeleton. Very few previous studies have attempted to measure this influence. Just one relevant study was found related to workplace exoskeletons. Tan et al. [7] studied synergies while performing tasks with exoskeletons. We also found that muscle coordination differed when using the exoskeleton. In particular, the coordination between muscle pairs was higher when the exoskeleton was used (higher MI, lower MSE, and higher CORR).

The study of coordination and synergies is more common among rehabilitation exoskeletons and assistive exoskeletons, but the purpose is different to that of our study. For rehabilitation purposes, in some pathologies derived from brain injuries, the process of patient recovery involves plastic neural re-wiring, establishing in some cases pathological synergies that should be avoided to allow more functional movements [46,47,48].

The MSE exhibited remarkable consistency among almost all pairs of muscles. Multivariate entropy showed high variations when the exoskeleton was not used, but these variations became slight when the exoskeleton was employed, resulting in a very flat slope during the performance of the exercises, showing efforts to maintain the muscular relationship with the exoskeleton. Similar to other studies that have utilized this parameter to characterize muscle pair synchronization (e.g., [19,21,22]), lower values were associated with a higher degree of coupling, and in the present case, this coupling was influenced by the action of the exoskeleton.

The results obtained for MSE demonstrated its suitability for studying couplings between pairs of muscles in this application. However, the remaining variables proved to be important for providing context for the information, partly because the MSE was more challenging to interpret on its own, as it conveyed information related to signal predictability and relationships.

Both the CORR and COH indicated very weak relationships between the considered muscles, with values around 0.2. Only the COH between the quadriceps and gluteus was somewhat higher (0.4), showing a closer causal relationship between these muscles.

The COH, MI, and MSE variables revealed that the exoskeleton tended to maintain the degree of coupling throughout the exercise. This led us to think that the use of the exoskeleton restricted the degrees of freedom of movement and the harmonization of the redistribution of loads. Some muscles that fatigued without the exoskeleton did not fatigue with the exo, and this displacement of the spectrum towards low frequencies no longer occurred, therefore reducing the coherence (the linear similarity of their spectra decreased).

This study stands as an initial approach to characterizing the phenomena of coupling differences between the conditions of wearing and not wearing an exoskeleton. Further studies are required for a deeper understanding, including the design of a methodology specifically for the study of couplings with more superficial muscles and pairs of antagonists. Also, the study of synergies through NNMF would be of great interest.

## 5. Limitations and Future Work

The main limitations of the present work lie in the fact that we analyzed a specific task, although common in industrial environments, with a single commercial exoskeleton, which limited the generalizability of the results obtained. Furthermore, only a small set of muscles were analysed in the study.

In our study, the tasks conducted with the users wearing the exoskeleton were always performed after the tasks without the exoskeleton. This sequencing choice was made for instrumentation purposes, as the removal of the exoskeleton could potentially cause slight skin displacements of the EMG electrodes, complicating comparisons between conditions. While acknowledging that this decision might have had an impact on the analysis of the evolution of muscle activation, we believe its influence was minimal: the observed trend when using the exoskeleton was largely contrary to the pattern identified without its use, underscoring our confidence that any potential impact had little effect on the study’s conclusions.

The consequences of the differences in muscle coordination when using or not using the exoskeleton, i.e., whether they imply advantages or drawbacks, remain a subject for further study. There is ample opportunity for investigating the long-term health effects on users.

It is worth noting that the study primarily focused on the objective evaluation of tasks with short durations and limited movement types. To comprehensively assess acceptance and long-term effects, a longitudinal study encompassing a broader range of tasks will be undertaken.

Furthermore, in the realm of future research, there are plans to explore alternative models of passive lumbar exoskeletons to validate the obtained results. Additionally, considering that one of the primary goals of occupational exoskeletons is a reduction in musculoskeletal risks, it would be interesting to investigate the relationship between the acquired parameters and their impact on musculoskeletal health.

## 6. Conclusions

The findings suggested that the use of a lumbar exoskeleton had an impact on the synchronization of the monitored muscle pairs.

A notable decrease in the values of the MSE parameter, along with a general reduction in the slope, was observed in all muscle pairs analyzed, except for the LUMB-SEMI pair, which indicated an increase in synchronization when the exoskeleton was utilized. This was supported by the increased MI values, with statistically significant differences found for the GLUT-LUMB pair. The results for the CORR and COH, which assessed the linear relationships between muscle pairs in the temporal and spectral domains, were less pronounced.

The MSE proved to be a highly sensitive variable for identifying differences in muscle coordination associated with exoskeleton use. While the results for the MSE and MI were consistent with each other, more significant differences were identified with the MSE.

DGC provided insights into the changes in coordination as the task progressed and could be a valuable tool for understanding how the users adjusted their strategy. In our case, DCG revealed changes in coordination in the “without exoskeleton” condition, while the coordination aspects remained relatively stable in the “with exoskeleton” condition.

Finally, as far as we are concerned, the changes in the directionality of muscular coupling associated with the use of exoskeletons were a novel finding of the present work and could provide a new tool in the assessment of occupational exoskeletons, allowing one to predict the effects of medium- and long-term usage.

## Figures and Tables

**Figure 1 sensors-23-09631-f001:**
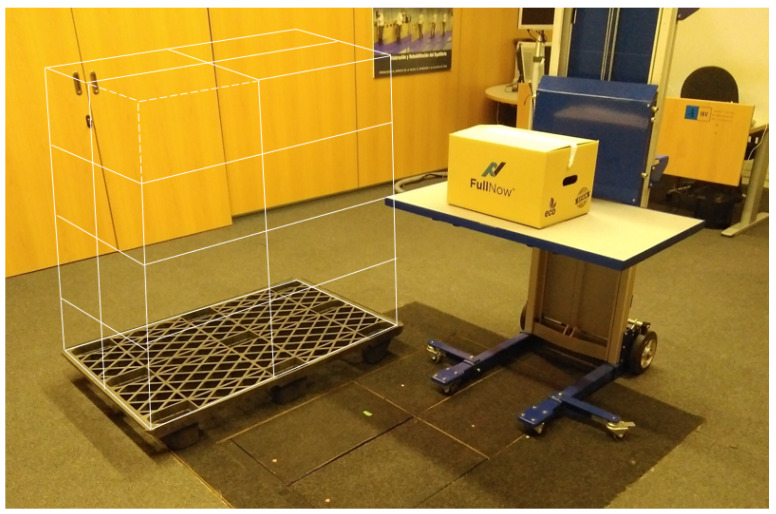
Picture of the laboratory configuration, showing the box over the destination table, and a schematic drawing in white of the initial configuration of the 16-box pallet.

**Figure 2 sensors-23-09631-f002:**
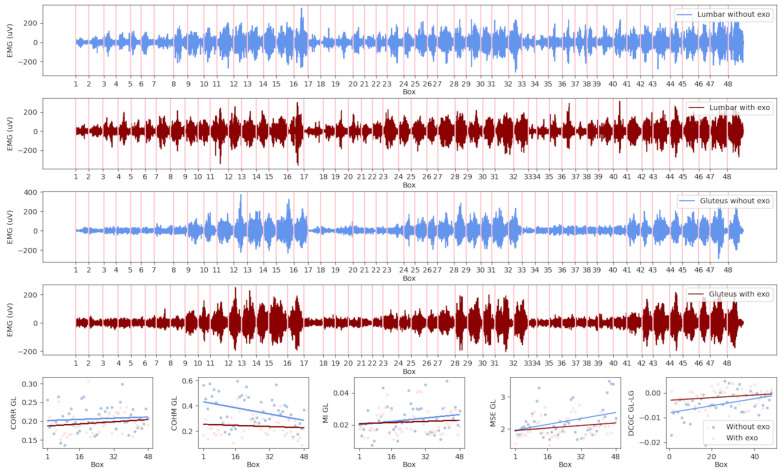
Subplots containing an example of a user’s signals and parameters calculated for the lumbar (LUMB) and gluteus (GLUT) pair of muscles. In the first four rows, the 48 fragments of the EMG signals for the lumbar and gluteus are concatenated (red—with exoskeleton, blue—without exoskeleton). The last row shows each of the five parameters calculated for the EMG segments of the GLUT-LUMB pair (light red dots—with exoskeleton, light blue dots—without exoskeleton). Over the dots, the lines of the trends are represented (red—with exoskeleton, blue—without exoskeleton).

**Figure 3 sensors-23-09631-f003:**
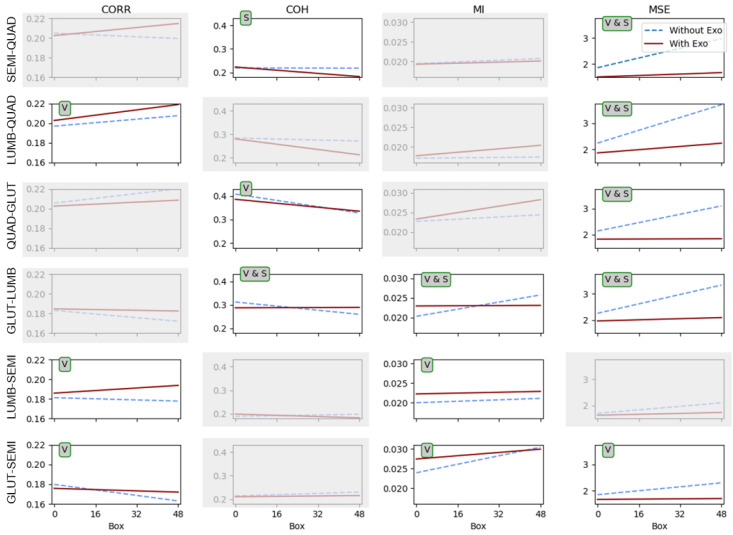
Plots of the trends for the conditions with (solid red lines) and without (dashed blue lines) the exoskeleton. The columns show each of the four parameters: CORR, COH, MI, and MSE. The rows show each of the six muscle combinations. “V”—significant differences in the values between conditions, “S”—significant differences in the slope between conditions, and “V & S”—significant differences in both the values and the slope.

**Figure 4 sensors-23-09631-f004:**
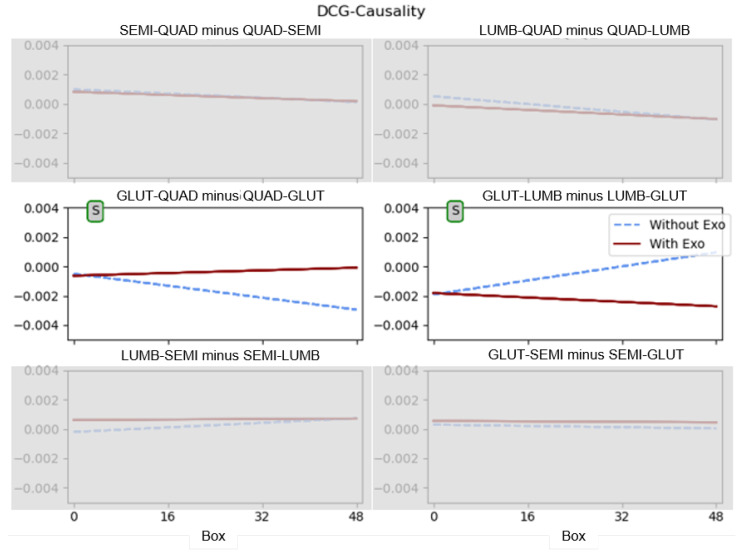
Matrix of representations of the slopes calculated for the DCG-Causality parameter under conditions with (solid red line) and without (dashed blue line) the exoskeleton. In the graphs, each of the six muscle combinations are shown as follows: semitendinosus to quadriceps minus quadriceps to semitendinosus (SEMI-QUAD minus QUAD-SEMI). “V”—significant differences in the values between conditions, “S”—significant differences in the slope between conditions, and “V & S”—significant differences in both the values and the slope.

**Table 1 sensors-23-09631-t001:** Matrix of *p*-values corresponding to the plots in Figure 3. Only significant differences are included, with values of p≤0.05.

	CORR		COH		MI		MSE	
	V	S	V	S	V	S	V	S
SEMI-QUAD	-	-	-	0.020	-	-	0.000	0.000
LUMB-QUAD	0.040	-	-	-	-	-	0.000	0.000
QUAD-GLUT	-	-	0.012	-	-	-	0.000	0.000
GLUT-LUMB	-	-	0.001	0.040	0.000	0.016	0.000	0.000
LUMB-SEMI	0.030	-	0.050	-	0.000	-	-	-
GLUT-SEMI	0.050	-	-	-	0.000	-	0.005	0.06

**Table 2 sensors-23-09631-t002:** Matrix of *p*-values corresponding to the plots in Figure 4; only significant differences included, with values of p≤0.05.

DCG-Causality		
	V	S
SEMI-QUAD minus QUAD-SEMI	-	-
LUMB-QUAD minus QUAD-LUMB	-	-
GLUT-QUAD minus QUAD-GLUT	-	0.04
GLUT-LUMB minus LUMB-GLUT	-	0.03
LUMB-SEMI minus SEMI-LUMB	-	-
GLUT-SEMI minus SEMI-GLUT	-	-

## Data Availability

Data are contained within the article.

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
