# Peer review of "Assessment of Muscle Coordination Changes Caused by the Use of an Occupational Passive Lumbar Exoskeleton in Laboratory Conditions"

_sensors, 2023, doi:10.3390/s23249631_

Round 1
Reviewer 1 Report
Comments and Suggestions for Authors
The paper is very focused on the metrics / calculation types that you used. So much detail is spent on discussion prior work that used Correlation Coefficient, Mutual Information, Multivariate Sample Entropy, and Conditional Granger Causality. Possibly due to lack of these being applied to similar activities as this paper the examples all seem very random and didn’t really lead me to care about the application. It was explaining the calculations but not motivating me on the work being presented. Similarly, the paper spends so much time going over the math of each method which might help a few people as a reference but probably the math has been published before or most researchers would just access the metric as a package in a stats program and not really need this level of detail.
The level of back ground about the metrics and the math (about 4 pages) doesn’t balance out well with the level of interpretation/discussion (1 page) which spends also too much time saying how there is nothing to compare to and a paragraph that is more intro style that summarizes some prior works but doesn’t connect them to the results or give interpretation.
I have no doubt that the math was done correctly and may be unique but I struggled to take much away on the application of those metrics to a problem. There was also a lack of order randomization between the conditions (exo – no exo) and a fairly limited number of muscles tested (just 4).
Line 31: More than one study should be cited to support this major assertion about prior “studies”. Suggest this even if you are citing a review article.
Line 33: Again you say “works” and cite only one study
Line 38: “most benefited one” is awkward or poor grammar
Lines 40: Some opening clauses don’t fit the rest of the sentence or are very informal ex: “on the other hand”
Lines 50-55: Should cite some works defining or using these metrics you list
Line 60 and 62 : should be “also obtaining” and “also reflect”
Lines 58 and 66: Seem to address the same point and same study
Line 68: We would normally say post-stroke or chronic stroke
Line 104: stay in past tense “processed them” also bad grammar in the same sentence
Line 129: Consisted OF
Line 135: Major experimental design issue is that the non exoskeleton tasks were always performed first and the exoskeleton were always performed second. Randomization is a very basic scientific best practice. You can’t rule out that differences due to non-exo vs exo weren’t in fact influenced by the order that the tasks were always performed in. The argument about “worst case fatigue” doesn’t fully make sense. Was there a break between and for how long?
Line 157: “a filtering stage” needs to be more clear. What filter types and what cut-off frequencies were used, rectification, ect in what order. This only takes a sentence to ensure that your methods are consistent with normal EMG processing methods and could be repeated by other researchers. Seniam recommends high pass around 10-20 Hz to remove motion artifact and DC issues, then lowpass near 500 Hz (200 Hz is at the very low end and should be supported with prior studies on similar muscles). Typically, then rectifying and then low pass filtering again to get the activation envelope.
Line 216: is built, not build
Line 228: the more, not the most
Line 260: depicts, not depicted and for, not por
Line 269: spelling
Line 281: again “on the one hand” ect is a colloquialism that sounds odd in a technical paper
Line 285: You use GL for Gluteus in the caption but later for Gluteus-Lumbar as a comparison pair. I found the acronyms for the pairs very hard to follow. For example Quadriceps – Gluteus would be much better as QUAD-GLUT if you want it shorter than QG which generally would make readers think you are referring to one muscle rather than a pairing. It just becomes draining to read because people can’t remember 6 random two letter combos over several more pages.
Discussion
As a rule, reviewers will immediately flag a paper that says there are “no other papers” to compare results to. You clearly already cite some papers in your discussion. Better to find some papers with common features and focus on what you can compare against in the prior literature. You may only get papers what match on some similar features to your work not all of them. Your paper features: with and without exo, back and leg muscle EMG data, loading task, fatigue, specific ways of measuring muscle couplings. You might find papers that match 4 of the 5 and should try to cite those. If you feel you are already doing this then I would take out the preface here about not having anything to compare to.
While you cite some other papers it’s more like a list (introduction style) rather than trying to connect your results to those studies. An effort should be made to address this and make some connections. Seems to only happen in line 396.
Possible paper suggestion: https://doi.org/10.1080/00140139.2020.1870162
Line 397: The one sentence paragraph should get connected to the next one
Line 390: Again everyone wants to feel that their work is completely novel but in reality this is extremely rarely the case and just comes across as being too narrow focused or too lazy when trying to find other literature. It’s much better to say that the prior work is “limited” and then explain exactly what you are adding to that prior literature (even if only a few papers) “we can understand a bit from paper A but have also added analysis X Y Z” ect.
Line 399: don’t think corroborate is the right word here since that means you are supporting a prior work
Limitations: You don’t have a limitation section. Strongly suggest to add this explicitly. No work is without limitations and most people expect to see this between the discussion and conclusions. For example only 4 muscles were tested, that’s a very low number for a study focused on EMG results.
Comments on the Quality of English Language
Paper needs to be proofread by a native English speaker. While 98% is perfect that last 2% added confusion and distraction for myself as a native speaker. I tried to note all the issues that were wrong/distracting but ideally a technical reviewer shouldn’t have to spend this much time addressing grammar / word choice issues.
Author Response
Line 31: More than one study should be cited to support this major assertion about prior “studies”. Suggest this even if you are citing a review article.
Line 33: Again you say “works” and cite only one study
Response to the referee and changes in the manuscript: Thank you for your comments, you are completely right. We have included more references in the revised manuscript according to the use of “plural”.
“Previous works related to the study of the effects of industrial passive exoskeletons on users have included the assessment of muscles through EMG [7,8] These studies focused on quantifying muscular activity and fatigue, but they only considered the effects on each muscle individually.”
Line 38: “most benefited one” is awkward or poor grammar
Response to the referee and changes in the manuscript: Thank you, for your observation, we have modified this sentence in the revised versión of the manuscript as follows
Lines 40: Some opening clauses don’t fit the rest of the sentence or are very informal ex: “on the other hand”
Response to the referee: Thank you for your comment. We have thoroughly reviewed the entire document to avoid the informal use of the English language.
Lines 50-55: Should cite some works defining or using these metrics you list
Response to the referee and changes in the manuscript: Thank you for this remark. We have included two references in this regard in the revised manuscript:
“One of the most common ways to study the muscle couplings in the literature consists of the calculation of certain parameters, such as the Correlation Coefficient, the Coherence Coefficient, the Mutual Information, and the Multivariate Sample Entropy [20,21]. “
The references are:
“20. Mas-Cabo, J.; Ye-Lin, Y.; Garcia-Casado, J.; Alberola-Rubio, J.; Perales, A.; Prats-Boluda, G. Uterine contractile efficiency indexes for labor prediction: A bivariate approach from multichannel electrohysterographic records. Biomedical Signal Processing and Control 2018, 46, 238–248. doi:10.1016/j.bspc.2018.07.018.
- Ye-Lin, Y.; Prats-Boluda, G.; Galiano-Botella, M.; Roldan-Vasco, S.; Orozco-Duque, A.; Garcia-Casado, J. Directed Functional Coordination Analysis of Swallowing Muscles in Healthy and Dysphagic Subjects by Surface Electromyography. Sensors 2022, 22. doi:10.3390/s22124513.”
Line 60 and 62 : should be “also obtaining” and “also reflect”
Response to the referee and changes in the manuscript: Sorry for this mistake. We have fixed it as recommended
Lines 58 and 66: Seem to address the same point and same study (nueva línea 86)
Response to the referee and changes in the manuscript
Line 68: We would normally say post-stroke or chronic stroke
Response to the referee and changes in the manuscript: Thank you for your remark. We have changed it accordingly:
“Coherence was computed to study post-stroke muscle interactions in Deltoid and Triceps [25], the values of coherence were lower for patients than for control.”
Line 104: stay in past tense “processed them” also bad grammar in the same sentence
Response to the referee and changes in the manuscript: Sorry for the mistakes, We have revised the use of English throughout the manuscript. Specifically, we have rewritten all this paragraph.
Line 129: Consisted on
Response to the referee and changes in the manuscript: Sorry for this mistake, We have amended it and revised the use of English throughout the manuscript.
Line 135: Major experimental design issue is that the non exoskeleton tasks were always performed first and the exoskeleton were always performed second. Randomization is a very basic scientific best practice. You can’t rule out that differences due to non-exo vs exo weren’t in fact influenced by the order that the tasks were always performed in. The argument about “worst case fatigue” doesn’t fully make sense. Was there a break between and for how long?
Response to the referee: Thank you for your observation. As you point out, we have not carried out randomization in the use of the exoskeleton during the recording protocol. .The reason why the recording protocol was always initiated without the use of exoskeleton was essentially to prevent that the placement and removal of the exoskeleton could affect the recording sensors and therefore causing a bias in the measurement associated with their repositioning. On the other hand, the protocol contemplated a 10-minute break between the registration without an exoskeleton and with an exoskeleton, which, given that the duration of the execution of the depalletizing task lasted about 90 seconds, represented more than 6 times the execution time and therefore enough for recovering if during the execution of the exercise without exoskeleton the subject has suffered from fatigue.
Changes in the manuscript:
The sentence has been written and the expression "worst case" has been removed to avoid misunderstandings. We have included in material and methods section the duration of the break between tasks. Furthermore, it has been included a section, “Limitations and future work where we have discussed why we always started the recording protocol without exoskeleton and we did not carry out randomization in the use of the exoskeleton during the recording protocol.
Line 157: “a filtering stage” needs to be more clear. What filter types and what cut-off frequencies were used, rectification, ect in what order. This only takes a sentence to ensure that your methods are consistent with normal EMG processing methods and could be repeated by other researchers. Seniam recommends high pass around 10-20 Hz to remove motion artifact and DC issues, then lowpass near 500 Hz (200 Hz is at the very low end and should be supported with prior studies on similar muscles). Typically, then rectifying and then low pass filtering again to get the activation envelope.
Response to the referee and changes in the manuscript: Thanks for your comment. Additional explanations have been included in materials and methods section to clarify the steps carried out in the preprocessing of the signals:
“EMG envelop has been obtained rectifying and lowpass filtering the rectified signal by a 4th order Butterworth filter with a cut-off filtering of 4 Hz. The envelope was used to identify the activation segments in the sEMG signal.”
Furthermore, the primary reason for filtering the sEMG signals below 200Hz was that in a preliminary study conducted by the research group, it was observed that the spectral content of these signals was mainly concentrated below 200Hz, with the contribution beyond this frequency being marginal. Therefore, to save computational cost, filtering and down-sampling of the signals below 200Hz was decided.
Line 216: is built, not build
Line 228: the more, not the most
Line 260: depicts, not depicted, and for, not por
Line 269: spelling
Line 281: again “on the one hand” ect is a colloquialism that sounds odd in a technical paper
Response to the referee and changes in the manuscript: Sorry for these mistakes, We have revised the use of English throughout the manuscript and avoided non-technical language
Line 285: You use GL for Gluteus in the caption but later for Gluteus-Lumbar as a comparison pair. I found the acronyms for the pairs very hard to follow. For example Quadriceps – Gluteus would be much better as QUAD-GLUT if you want it shorter than QG which generally would make readers think you are referring to one muscle rather than a pairing. It just becomes draining to read because people can’t remember 6 random two-letter combos over several more pages.
Response to the referee and changes in the manuscript: Thank you for this remark. As you have indicated, the acronyms used in the original manuscript could be misleading. We have changed the acronyms as suggested throughout the manuscript to facilitate the interpretability of the figures and the tracking of the text's reading.
Discussion
As a rule, reviewers will immediately flag a paper that says there are “no other papers” to compare results to. You clearly already cite some papers in your discussion. Better to find some papers with common features and focus on what you can compare against in the prior literature. You may only get papers what match on some similar features to your work not all of them. Your paper features: with and without exo, back and leg muscle EMG data, loading task, fatigue, and specific ways of measuring muscle couplings. You might find papers that match 4 of the 5 and should try to cite those. If you feel you are already doing this then I would take out the preface here about not having anything to compare to.
While you cite some other papers it’s more like a list (introduction style) rather than trying to connect your results to those studies. An effort should be made to address this and make some connections. Seems to only happen in line 396.
Possible paper suggestion: https://doi.org/10.1080/00140139.2020.1870162
Line 397: The one sentence paragraph should get connected to the next one
Line 390: Again everyone wants to feel that their work is completely novel but in reality this is extremely rarely the case and just comes across as being too narrow focused or too lazy when trying to find other literature. It’s much better to say that the prior work is “limited” and then explain exactly what you are adding to that prior literature (even if only a few papers) “we can understand a bit from paper A but have also added analysis X Y Z” ect.
Line 399: don’t think corroborate is the right word here since that means you are supporting a prior work
Response to the referee and changes in the manuscript: The discusión has been thoroughly revised and modified considering all above mentioned aspects. We have focused on the interpretation of the results and their implications in what concerns the aspects of muscular coordination related to exoskeletons including references in this regard as you suggested.
Limitations: You don’t have a limitation section. Strongly suggest to add this explicitly. No work is without limitations and most people expect to see this between the discussion and conclusions. For example only 4 muscles were tested, that’s a very low number for a study focused on EMG results.
Response to the referee and changes in the manuscript: Thank you for this comment, we appreciate it. We have added a new section in the revised manuscript regarding limitations and future lines.
Paper needs to be proofread by a native English speaker. While 98% is perfect that last 2% added confusion and distraction for myself as a native speaker. I tried to note all the issues that were wrong/distracting but ideally a technical reviewer shouldn’t have to spend this much time addressing grammar / word choice issues.
Response to the referee and changes in the manuscript: You are completely right. The revised manuscript has been revised by a native English speaker expert in technical papers revision.
Reviewer 2 Report
Comments and Suggestions for Authors
Dear authors, thank you for submitting the document entitled “Effects of the use of a passive lumbar exoskeleton in muscle under laboratory conditions”. It is very interesting and nicely written. Aiming to improve the overall quality of the document, some amendments are suggested.
1. Please consider making the title more informative of what has been done, including information regarding the population, the intervention, the comparison group, and the outcome being measured.
2. In the abstract, results section, please include a description of the most prominent results, namely significance values, instead of describing all the statistical analyses performed.
3. In the introduction section, lines 35 and 36 “The study of the effects over individual muscles carried out by other authors start 35 with the selection of objective muscles.”, please consider rewriting this sentence for clarity. Line 102 and 103, please consider rewriting this sentence for clarity.
4. In the methods section, please describe the rationale for studying the following muscles: erector spinae, gluteus medius, quadriceps femoris, and semitendinosus. Furthermore, why did you choose to not filter the powerline noise? Please consider describing skin preparation for the reduction of impedance. What was the rationale for the reduction of movement noise in a slow-motion task?
5. In the results section, table 1, please consider the use of a power of 10 with a negative exponent.
6. In the discussion section please consider removing personal opinions like “Encountering situations where there are no other papers to compare results can be both challenging and thought-provoking.” Please consider moving the following information “Further studies would be required for a deeper understanding, involving designing the methodology specifically for the study of couplings with more superficial muscles and by pairs of antagonists. Also, the study of synergies through the NNMF would be of great interest.”, which are directions for future studies, to the end of the conclusions section. At the end of the discussion section please include all the limitations of your study.
7. In the conclusion section please consider writing “exoskeleton” in the extended form instead of “exo” (line 428).
Comments on the Quality of English LanguageThe document is nicely written.
Author Response
Dear authors, thank you for submitting the document entitled “Effects of the use of a passive lumbar exoskeleton in muscle under laboratory conditions”. It is very interesting and nicely written. Aiming to improve the overall quality of the document, some amendments are suggested.
- Please consider making the title more informative of what has been done, including information regarding the population, the intervention, the comparison group, and the outcome being measured.
Response to the referee and changes in the manuscript: We appreciate your comment and we have changed the title to “Assessment of muscle coordination changes caused by the use of an occupational passive lumbar exoskeleton in laboratory conditions”. This new title tries to answer all the concerns raised about the objective and application of the work.
- In the abstract, results section, please include a description of the most prominent results, namely significance values, instead of describing all the statistical analyses performed.
Response to the referee and changes in the manuscript: Thank you for this remark. We have modified the abstract attending to your indications.
- In the introduction section, lines 35 and 36 “The study of the effects over individual muscles carried out by other authors start 35 with the selection of objective muscles.”, please consider rewriting this sentence for clarity. Line 102 and 103, please consider rewriting this sentence for clarity.
Response to the referee and changes in the manuscript: Thank your comments. We have rewritten:
"The study starts with the identification of the muscles affected by the use of the exoskeleton. In the case of the passive lumbar exoskeleton, the lumbar muscle is selected for being the objective muscle of the device, as it should be the potentially most benefited one"
The paragraph related to lines 102 and 103 has been also rewriten.
- In the methods section, please describe the rationale for studying the following muscles: erector spinae, gluteus medius, quadriceps femoris, and semitendinosus. Furthermore, why did you choose to not filter the powerline noise? Please consider describing skin preparation for the reduction of impedance. What was the rationale for the reduction of movement noise in a slow-motion task?
Response to the referee and changes in the manuscript: Thank you for your observation. We decided to record erector spinae, gluteus medius, quadriceps femoris, and semitendinosus muscles because they are muscles directly involved in task execution and can be affected by the use of the lumbar exoskeleton. In the future, we aim to record a greater number of muscles to obtain a more comprehensive assessment of the exoskeleton's effect on a larger number of muscle groups and, consequently, on the overall musculature, as included in a new section of the revised manuscript “Limitations and future works and practical implications”.
As for the powerline noise removal it was not carried out to preserve the original"I computed the Fast Fourier Transform (FFT) of the muscle activation segments, and in the periodograms, there was no apparent presence of power line interference, indicating a high Signal-to-Noise Ratio (SNR) during muscle activation. Please note that the recordings were conducted in a laboratory, in a controlled environment. Additionally, it's worth mentioning that a couple of recordings were discarded due to the presence of motion artifacts
About skin preparation, the skin area where the electrodes were placed was carefully cleaned, exfoliated, and wiped with alcohol. This information has been added in the revised manuscript in material and methods section.
Finally, regarding motion artifacts, these occur due to the movement/displacement that takes place between the electrodes and the skin, and they persist even when the movements are slow.
- In the results section, table 1, please consider the use of a power of 10 with a negative exponent.
Response to the referee and changes in the manuscript: Typically, p-values from an ANOVA are represented in decimal notation as it is considered a level of statistical significance p<0.05. Only values with statistical significance (p<0.05) are represented in table 1.
- In the discussion section please consider removing personal opinions like “Encountering situations where there are no other papers to compare results can be both challenging and thought-provoking.” Please consider moving the following information “Further studies would be required for a deeper understanding, involving designing the methodology specifically for the study of couplings with more superficial muscles and by pairs of antagonists. Also, the study of synergies through the NNMF would be of great interest.”, which are directions for future studies, to the end of the conclusions section. At the end of the discussion section please include all the limitations of your study.
Response to the referee and changes in the manuscript: We appreciate your remarks and truly believe that your comments have allowed us to enhance the quality of the manuscript. In the revised version of the manuscript, we have rewritten the discussion, avoiding the inclusion of personal opinions. Likewise, we have added a new section on limitations and future studies.
- In the conclusion section please consider writing “exoskeleton” in the extended form instead of “exo” (line 428).
Response to the referee and changes in the manuscript: Thank you for your observation. We have changed it in the revised manuscript
Round 2
Reviewer 1 Report
Comments and Suggestions for Authors
163: Many issues with this sentence spelling and English. The low pass cutoff of 4 Hz is very low, should reference another paper that uses this cutoff or recommends it for lifting tasks.
169: Can be found
314: Maybe should be SEMI-QUAD ect
Discussion:
Still feel that the authors are not doing much to connect this work to prior work on exoskeletons. They only cite one other exoskeleton paper in the whole discussion and it is from the same group (the other 3 papers cited relate to the methods not exoskeletons). While the discussion is no longer making claims of there being no other work to compare it to I feel it’s quite low effort to literally not look for a single paper beyond your group that examines exoskeleton EMG, exoskeleton fatigue, exoskeleton lifting tasks, or just lifting tasks alone ect.
Limitations:
Author Response:
Changes in the manuscript:
The sentence has been written and the expression "worst case" has been removed to avoid misunderstandings. We have included in material and methods section the duration of the break between tasks. Furthermore, it has been included a section, “Limitations and future work where we have discussed why we always started the recording protocol without exoskeleton and we did not carry out randomization in the use of the exoskeleton during the recording protocol.
I do not see where this was added in the Limitations.
Comments on the Quality of English Languagesome language and spelling issues remain, see other comments
Author Response
163: Many issues with this sentence spelling and English. The low pass cutoff of 4 Hz is very low, should reference another paper that uses this cutoff or recommends it for lifting tasks.
Author Response:
We totally agree with your comment. The 4 Hz low pass filter is not for pre-processing the EMG signals but for the automatic segmentation of the EMG activations, which is explained in the second paragraph of section 2.3.1. The cut-off frequencies for pre-processing EMG signals were set between 20 Hz and 200 Hz, as mentioned in the first paragraph of that section. We have changed that sentence from the first paragraph to the second one in the revised manuscript, where assisted segmentation is described and referenced. Furthermore, the sentence has been slightly modified to correct the wording.
Changes in the manuscript:
Lines 163 and 164, “EMG envelop has been obtaining rectifying and lowpass filtering the rectified signal by a 4th order Butterworth filter with a cut-off filtering of 4 Hz”, have been removed.
Lines 165 to 169 have been modified to include the previous sentence, as follows: “Secondly, a segmentation task for selecting the fragments of muscular signal activation when the user was holding the box, from lifting to downloading, common to all muscle channels was carried out. In total, 48 fragments (three exercises of 16 boxes each) for each muscle in both, with and without exoskeleton condition. The extended details of the segmentation methodology can be found in a previous work [9]; which is based on calculating the envelope of the EMG signals in the four muscles, by rectifying and smoothing with a 4 Hz lowpass filter.”
Comments and Suggestions for Authors
169: Can be found
314: Maybe should be SEMI-QUAD ect
Author Response:
Thank you for pointing out these mistakes. Both have been corrected in the manuscript, and we have also changed the muscle pair “Gluteus-Lumbar” to “GLUT-LUMB” in the same line (314).
Changes in the manuscript
Line 169, change “find” to “found”
Line 314, change “…. muscles of Semitendinosus-Quadriceps and Gluteus-Lumbar.” to “….. muscles of SEMI-QUAD and GLUT-LUMB.”
Discussion:
Still feel that the authors are not doing much to connect this work to prior work on exoskeletons. They only cite one other exoskeleton paper in the whole discussion and it is from the same group (the other 3 papers cited relate to the methods not exoskeletons). While the discussion is no longer making claims of there being no other work to compare it to I feel it’s quite low effort to literally not look for a single paper beyond your group that examines exoskeleton EMG, exoskeleton fatigue, exoskeleton lifting tasks, or just lifting tasks alone ect.
Author response:
We apologize if, in the previous response, we gave you the impression that we did not address your suggestion. We agree that many of the references mentioned in the discussion section of the previous manuscript referred to works from our research group, although not all of them as follows:
Tan, C.K.; Kadone, H.; Miura, K.; Abe, T.; Koda, M.; Yamazaki, M.; Sankai, Y.; Suzuki, K. Muscle synergies during repetitive stop lifting with a bioelectrically-controlled lumbar support exoskeleton. Frontiers in Human Neuroscience 2019, 13, 142.
Zhang, Y.; Hao, D.; Yang, L.; Zhou, X.; Ye-Lin, Y.; Yang, Y. Assessment of Features between Multichannel Electrohysterogram for Differentiation of Labors. Sensors 2022, 22. doi:10.3390/s22093352
As you indicate, there are multiple studies related to surface electromyography and exoskeletons in the literature. Although many studies have been reported in the last years (Google Scholar found references on 28th November 2023 with the keywords Exoskeleton EMGWorkplace), most of them are related to the comparison of muscle activations with and without the exoskeleton (Gillette & Stephenson, 2018; Pinho et al., 2020; Schmalz et al., 2019) by computing temporal or spectral features of the EMG signals from individual muscles. Similarly, the impact on muscle fatigue derived from the use of exoskeletons has been addressed by working on temporal and spectral features in target muscles, but not considering changes in their coupling (Gillette et al., 2022; Z et al., 2021).
As for the analysis of coordination and synergies, it is prevalent in rehabilitation and assistive exoskeletons in literature, but the objective differs in our case. In rehabilitation for certain pathologies resulting from brain injuries, the patient recovery process entails plastic neural rewiring, leading to the establishment of pathological synergies in some cases. These synergies should be avoided to facilitate more functional movements (Hassan et al., 2018; He et al., 2021; Tan et al., 2020). In the present work, we focus on identifying variations in couplings and interactions associated with the use of a lumbar exoskeleton in target muscles, and this is precisely the main contribution of the present study. Few prior investigations have endeavored to gauge this impact. Only a relevant study has been identified concerning workplace exoskeletons, where Tan et al. (Tan et al., 2019) explored synergies during the execution of tasks with exoskeletons.
Changes in the manuscript:
We have expanded the discussion on the points highlighted by the reviewer regarding the use of electromyography associated with the use of occupational exoskeletons or in other settings. This includes a discussion on the type of analysis or assessments conducted and the contribution or focus of the present study in that context. Specifically, we have changed the first paragraph of the discussion section pointing out that the literature mainly focuses on the análisis of the effect of exoskeletons comparing muscular activations individually, including 3 references in this regard (Gillette & Stephenson, 2018; Pinho et al., 2020; Schmalz et al., 2019).
As for studies regarding the use of sEMG in assessing muscle fatigue and the use of exoskeleton, we have added the following sentence in lines 336 and 337 (and 2 references):
(Gillette et al., 2022; Z et al., 2021).
We have remarked in the discussion section, lines 344-346, that very few works focused on the variations in muscle coupling due to the use of exoskeletons, apart from the relevant work of Tan et al (Tan et al., 2019).
We have also added a paragraph (lines 360-363) in the discussion of the revised manuscript remarking that muscular coordination and synergy studies are more prevalent in rehabilitation and assistive exoskeletons. Still, the purpose differs in our case as follows:
“The study of coordination and synergies are more common among rehabilitation
exoskeletons and assistive exoskeletons, but the purpose is different as in our case. For
rehabilitation purposes in some pathologies derived from brain injuries, the process of patient recovery involves the plastic neural re-wiring establishing in some cases pathological
synergies that should be avoided to allow more functional movements [47–49].”
The sentence has been written and the expression "worst case" has been removed to avoid misunderstandings. We have included in the material and methods section the duration of the break between tasks. Furthermore, it has included a section, “Limitations and future work where we have discussed why we always started the recording protocol without an exoskeleton and we did not carry out randomization in the use of the exoskeleton during the recording protocol.
I do not see where this was added in the Limitations.
Author response: Sorry for this mistake, we did not submit the last version of the manuscript and that information was missing. In the new revised version of the manuscript, it can be found.
Changes in the manuscript:
“worst case” was removed. We have added in the material and methods section the break time between tasks (lines 140-143). Additionally, we have incorporated a section titled "Limitations and Future Work," where we explain the rationale behind commencing the recording protocol without the exoskeleton and the decision not to implement randomization in the exoskeleton usage throughout the recording protocol (lines 388-395).
Comments on the Quality of English Language some language and spelling issues remain, see other comments
Author response: Thank you for your observation. We have thoroughly reviewed the use of English throughout the manuscript.
References
Gillette, J. C., Saadat, S., & Butler, T. (2022). Electromyography-based fatigue assessment of an upper body exoskeleton during automotive assembly. Wearable Technologies, 3. https://doi.org/10.1017/wtc.2022.20
Gillette, J. C., & Stephenson, M. L. (2018). EMG analysis of an upper body exoskeleton during automotive assembly. Proceedings of the 42nd Annual Meeting of the American Society of Biomechanics.
Hassan, M., Kadone, H., Ueno, T., Hada, Y., Sankai, Y., & Suzuki, K. (2018). Feasibility of synergy-based exoskeleton robot control in hemiplegia. IEEE Transactions on Neural Systems and Rehabilitation Engineering, 26(6). https://doi.org/10.1109/TNSRE.2018.2832657
He, C., Xiong, C. H., Chen, Z. J., Fan, W., Huang, X. L., & Fu, C. (2021). Preliminary Assessment of a Postural Synergy-Based Exoskeleton for Post-Stroke Upper Limb Rehabilitation. IEEE Transactions on Neural Systems and Rehabilitation Engineering, 29. https://doi.org/10.1109/TNSRE.2021.3107376
Pinho, J. P., Parik Americano, P., Taira, C., Pereira, W., Caparroz, E., & Forner-Cordero, A. (2020). Shoulder muscles electromyographic responses in automotive workers wearing a commercial exoskeleton. Proceedings of the Annual International Conference of the IEEE Engineering in Medicine and Biology Society, EMBS, 2020-July. https://doi.org/10.1109/EMBC44109.2020.9175895
Schmalz, T., Schändlinger, J., Schuler, M., Bornmann, J., Schirrmeister, B., Kannenberg, A., & Ernst, M. (2019). Biomechanical and metabolic effectiveness of an industrial exoskeleton for overhead work. International Journal of Environmental Research and Public Health, 16(23). https://doi.org/10.3390/ijerph16234792
Tan, C. K., Kadone, H., Miura, K., Abe, T., Koda, M., Yamazaki, M., Sankai, Y., & Suzuki, K. (2019). Muscle synergies during repetitive stoop lifting with a bioelectrically-controlled lumbar support exoskeleton. Frontiers in Human Neuroscience, 13. https://doi.org/10.3389/fnhum.2019.00142
Tan, C. K., Kadone, H., Watanabe, H., Marushima, A., Hada, Y., Yamazaki, M., Sankai, Y., Matsumura, A., & Suzuki, K. (2020). Differences in Muscle Synergy Symmetry Between Subacute Post-stroke Patients With Bioelectrically-Controlled Exoskeleton Gait Training and Conventional Gait Training. Frontiers in Bioengineering and Biotechnology, 8. https://doi.org/10.3389/fbioe.2020.00770
Z, W., X, W., Y, Z., C, C., S, L., Y, L., A, P., & Y, M. (2021). A Semi-active Exoskeleton Based on EMGs Reduces Muscle Fatigue When Squatting. Frontiers in Neurorobotics, 15. https://doi.org/10.3389/FNBOT.2021.625479
Reviewer 2 Report
Comments and Suggestions for Authors
Dear authors, thank you for providing the revised version of your manuscript. Please consider rewriting the following sentence, since it is difficult to understand: "The recording protocol contemplated a 10-minute break between the registration and with exoskeleton, more than 6 times the execution time which is about 60s, for recovering if during the execution of the exercise without exoskeleton the subject has suffered from fatigue."
Furthermore, all the limitations should be described at the end of the discussion section, not after the conclusions.
kind regards
Comments on the Quality of English LanguageNeed revision.
Author Response
Dear authors, thank you for providing the revised version of your manuscript. Please consider rewriting the following sentence, since it is difficult to understand: "The recording protocol contemplated a 10-minute break between the registration and with exoskeleton, more than 6 times the execution time which is about 60s, for recovering if during the execution of the exercise without exoskeleton the subject has suffered from fatigue."
Author response and changes in the manuscript: Thank you for your observation. We have rewritten this sentence in the revised manuscript to clarify its meaning as follows: "The recording protocol included a 10-minute break between sessions with and without the exoskeleton. This break, exceeding 6 times the exercise execution time of approximately 60 seconds, allowed for recovery in case the subject experienced fatigue during the exercise without the exoskeleton.
Furthermore, all the limitations should be described at the end of the discussion section, not after the conclusions.
Author response and changes in the manuscript: Thank you for this remark. We have moved the limitations and future work at the end of the discussion section, before the conclusions.